# Partially View-aligned Clustering

**Zhenyu Huang**
College of Computer Science
Sichuan University, China
zyhuang.gm@gmail.com

**Peng Hu**
I2R
A*STAR, Singapore
penghu.ml@gmail.com

**Joey Tianyi Zhou**
iHPC
A*STAR, Singapore
joey.tianyi.zhou@gmail.com

**Jiancheng Lv**
College of Computer Science
Sichuan University, China
lvjiancheng@scu.edu.cn

**Xi Peng**[*]
College of Computer Science
Sichuan University, China
pengx.gm@gmail.com

## Abstract

In this paper, we study one challenging issue in multi-view data clustering. To be specific, for two data matrices $\mathbf{X}^{(1)}$ and $\mathbf{X}^{(2)}$ corresponding to two views, we do not assume that $\mathbf{X}^{(1)}$ and $\mathbf{X}^{(2)}$ are fully aligned in row-wise. Instead, we assume that only a small portion of the matrices has established the correspondence in advance. Such a partially view-aligned problem (PVP) could lead to the intensive labor of capturing or establishing the aligned multi-view data, which has less been touched so far to the best of our knowledge. To solve this practical and challenging problem, we propose a novel multi-view clustering method termed partially view-aligned clustering (PVC). To be specific, PVC proposes to use a differentiable surrogate of the non-differentiable Hungarian algorithm and recasts it as a pluggable module. As a result, the category-level correspondence of the unaligned data could be established in a latent space learned by a neural network, while learning a common space across different views using the "aligned" data. Extensive experimental results show promising results of our method in clustering partially view-aligned data.

## 1 Introduction

As one of the most important unsupervised technologies, data clustering has attracted much attention in recent years [22, 16, 6, 34]. Since most of the real-world data are presented in multiple views or modals, it is highly expected to explore and exploit the correlation and invariance across different views for data analysis [33, 30, 14, 27, 41].

In general, most existing multi-view clustering (MVC) approaches jointly learn a common representation to bridge the gap among different views and then achieve clustering using the common representation. The success of such a learning paradigm highly relies on a "well-established" dataset which has to satisfy two assumptions: 1) *completeness of data*: It requires that all examples appear in all views. Taking two view matrices $\mathbf{V}^{(1)}$ and $\mathbf{V}^{(2)}$ as a showcase, it assumes that $\mathbf{V}^{(1)}$ and $\mathbf{V}^{(2)}$ are with the same number of rows, where each row denotes a data point; 2) *correspondence of views*: It requires that $\mathbf{V}^{(1)}$ and $\mathbf{V}^{(2)}$ have the corrected correspondence in row-wise. In other words, $\mathbf{V}^{(1)}$ and $\mathbf{V}^{(2)}$ are fully aligned in advance. With the above two assumptions, the correlation and correspondence of multi-view data are available, thus making learning the common representation and clustering possible.

---

[*]Corresponding author.

Based on the above two assumptions, a variety of MVC methods [30, 14, 19, 23, 39, 37, 25] have achieved promising performance. In practice, however, it is a daunting task to collect the complete and fully-aligned multi-view data due to the complexity and discordancy in time and space. In other words, these existing works probably fail whether the data is with *partially data-missing problem* (PDP) or *partially view-aligned problem* (PVP). More specifically, PDP assumes that all views would miss some data and therefore results in many partial examples, *i.e.*, examples with some views missing. Recently, some works have attempted to solve this challenging problem [9, 31, 28].

In this paper, we focus on the solution of PVP rather than PDP. To the best of our knowledge, there are few efforts towards solving PVP in MVC so far. In PVP, only a portion of data is aligned across different views. Formally, for a given dataset $\{\mathbf{X}^{(v)}\}_{v=1}^{m} = \{\mathbf{A}^{(v)}, \mathbf{U}^{(v)}\}_{v=1}^{m}$, only $\{\mathbf{A}^{(v)}\}_{v=1}^{m}$ are aligned with correct correspondence while the correspondence of $\{\mathbf{U}^{(v)}\}_{v=1}^{m}$ is unknown. Here, $v$ denotes which view and $m$ denotes the total view number. A typical example of PVP is street surveillance as shown in Fig. 1. In the example, several cameras correspond to multiple views. Due to the inconsistency and complexity in time and space, an interest of object may appear in another monitor at different time $t_1$ with different position $p_1$, thus leading to the partially-aligned multi-view data. It should be pointed out that it is a daunting task to solve this problem due to the following three reasons. First, it is impossible to utilize the label to perform view alignment in the unsupervised setting such as clustering. Second, although the vanilla graph match methods such as Hungarian algorithm [10] could be used to seek the correspondence of views. However, it is impossible to plug it into a neural network due to the non-differentiable property of the Hungarian algorithm. Third, it is expected to jointly learn common representation and perform alignment into a unified framework so that the partially aligned information could be utilized to facilitate the multi-view clustering performance.

To this end, we propose possibly the first study on partially view-aligned clustering (PVC). To be exact, PVC establishes the correspondence of unaligned data with the help of the ground-truth aligned data, while learning a common representation by preserving the view-specific structure and cross-view consistency. The contributions of this work could be summarized as follows:

- We propose a new paradigm for multi-view clustering, termed partially view-aligned problem. The PVP is pervasive but ignored in a variety of real-world applications and our solution to this problem could alleviate even avoid the daunting assumption of *correspondence of views* in data collection.
- To tackle the partially view-aligned problem in MVC, we propose a novel neural network which simultaneously aligns a given partially aligned dataset in a latent space and learns the common representation across different views. The alignment module is a differentiable surrogate of the Hungarian algorithm, which could be plugged into any neural network to embrace the joint optimization with back-propagation. To the best of our knowledge, this could be the first effective deep solution which makes clustering partially view-aligned data possible.

## 2 Related work

Multi-view clustering methods aim to exploit the diverse and complementary information contained in different views [32], which could be roughly classified into three categories based on different formulations of view-specific similarity and cross-view consistency. Namely, multi-view canonical correlation clustering which utilizes the correlation among different views [29, 1, 30]; multi-view matrix decomposition clustering which exploits the mutual information based on the matrix factorization technology and then perform the clustering on the learned matrix with low-rank constraint [40]; and multi-view subspace clustering which jointly perform the subspace learning with view-specific similarity and learn the common space with cross-modal consistency[23, 36, 15]. Although the above approaches have achieved promising results in multi-view clustering, they highly rely on the aforementioned two assumptions, *i.e.*, "completeness of data" and "correspondence of views". Thus they are limited to handling the partially data-missing problem (PDP) and partially view-aligned problem (PVP) well.

Recently, there are some deep methods have been proposed to solve PDP [13, 28, 4, 9, 18, 17]. The basic idea of these works is to utilize the remained information in the available views to predict the

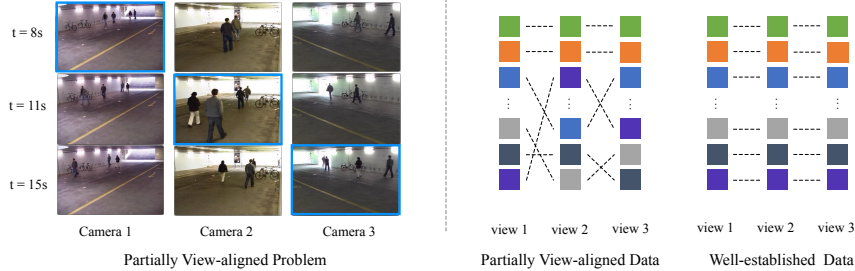

Figure 1: An illustrative example of the partially view-aligned problem. In the example, four individuals (clusters) walk through an underground passageway which is monitored by three cameras corresponding to three views. Clearly, it is a daunting task to collect the fully-aligned data based on either of the time and space positions. The blue-colored frames are a showcase of the "ideal" aligned data despite the space difference. On the right panel, we give a more direct illustration of the difference between the partially view-aligned data and well-established data. Different colors indicate different objects. In the example, only the first two rows are with the corrected correspondence.

missing samples. For example, [28] proposes using the paired views to train the networks so that the missing views are completed.

Compared to the PDP, PVP is more challenging and only a few works have tried to alleviate the effect caused by this problem. For example, [11] performs maximum covariance analysis (MCA) on the aligned data then progressively obtain the correspondence on the optimal cost matrix by using the existing Hungarian algorithm. [38] performs clustering for each view based on NMF and then utilizes the clustering result and the partially aligned information to establishes the correspondence of unaligned data. The major limitation of these methods is given below. On one hand, the methods are shallow models, and there is no efforts have been devoted to developing effective deep solution so far as we knew. On the other hand, these works establish the correspondence of views in a separate step. To utilize the representative capacity of the neural network, it is highly expected to jointly perform view alignment and the downstream task. As the view alignment is an NP-hard graph matching problem in essence, it is difficult to seek such a solution.

## 3    Partially View-aligned Clustering for Multi-view Data

In this section, we propose a deep multi-view clustering method, termed partially view-aligned clustering (PVC) which could solve the partially view-aligned problem as mentioned above. We first formally introduce the formulation of the partially view-aligned problem. Then we present how to tackle this problem with a detailed description of our proposed method based on a differentiable view alignment algorithm.

### 3.1    Problem Formulation

Given a dataset $\{\mathbf{X}^{(v)}\}_{v=1}^{m}$, MVC aims to separate all data points into one of $c$ clusters, where $\mathbf{X}^{(v)} = \{\mathbf{x}_1^{(v)}, \mathbf{x}_2^{(v)}, \cdots, \mathbf{x}_n^{(v)}\}^{\top}$. Furthermore, in our partially view-aligned setting, there only a part of $\{\mathbf{X}^{(v)}\}_{v=1}^{m}$ is with the correspondence in advance. To be specific, $\mathbf{X}^{(v)} = \{\mathbf{A}^{(v)}, \mathbf{U}^{(v)}\}$, where $\mathbf{A}^{(v)}$ and $\mathbf{U}^{(v)}$ denote the aligned data and unaligned data in row-wise for the $v$-th view.

To achieve clustering on the partially view-aligned data, one feasible way is to first establish the correspondence of the pair of views using the graph matching methods such as the Hungarian algorithm. After that, one performs multi-view clustering on the aligned data like the conventional pipeline. Although such a two-step paradigm is easy to follow, it has suffered from the following limitations. First, the vanilla Hungarian algorithm directly seeks the maximum-weight matchings in bipartite graphs, which cannot utilize the partially-aligned data. It is reasonable to believe that the performance would be facilitated if the partially-aligned data are exploited during alignment. Second, the Hungarian algorithm is non-differentiable, which cannot be plugged into a neural network. As a variety of studies [30, 40, 37] have shown the effectiveness of neural networks in MVC, it is highly expected to develop a differentiable alignment algorithm so that the common representation learning

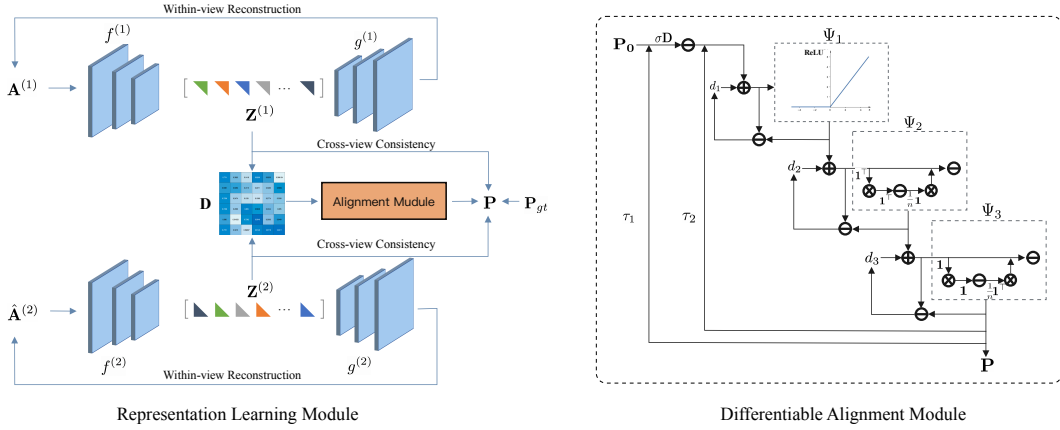

Representation Learning Module  Differentiable Alignment Module

Figure 2: Overview of the proposed method which consists of two modules, the representation learning module and the differentiable alignment module. The arrows show how we train the network with the aligned data $\mathbf{A}^{(1)}$ and shuffled $\hat{\mathbf{A}}^{(2)}$. We first obtain the latent representations $\mathbf{Z}^{(1)} = f^{(1)}(\mathbf{A}^{(2)})$ and $\mathbf{Z}^{(2)} = f^{(2)}(\hat{\mathbf{A}}^{(2)})$, then compute the distance matrix $\mathbf{D}$ on the hidden representations. Then, the permutation matrix $\mathbf{P}$ is computed by progressively feeding $\mathbf{D}$ into the alignment module. After that, we compute the loss to optimize the network.

and data alignment could be jointly optimized. The reasonability of such a joint learning paradigm relies on the following observations/assumptions. To be specific, the clustering performance will be improved by considering the correspondence/correlation at the set (*i.e.*, cluster) level during learning common representation. However, most of MVC methods only utilize the correspondence at the point level. Clearly, a pluggable alignment algorithm will alleviate such an issue and lead to a more desirable result.

Based on the above discussions, we aim to develop a neural network that could establish the correspondence of a given dataset using a differentiable surrogate of the Hungarian algorithm and simultaneously learn the common representation for different views by implicitly using the alignment information. Formally, one key step of our model is to seek the correspondence or so-called permutation matrix $\mathbf{P} \in \mathbb{R}^{n \times n}$ so that:

$$\mathbf{X}^{(1)} \sim \mathbf{P}\mathbf{X}^{(2)}, \tag{1}$$

where $\sim$ is a relational operator, which denotes that $\mathbf{X}^{(1)}$ and $\mathbf{X}^{(2)}$ are aligned with correct correspondence via $\mathbf{P}$. $\mathbf{P}$ is a square binary matrix that has exactly one entry of 1 in each row and column, and 0 elsewhere. In words, $\mathbf{P}$ will reorder the rows or columns while remaining the value of $\mathbf{X}^{(2)}$ unchanged.

As shown in Fig. 2, our PVC model consists of two modules, *i.e.*, one is to learn the cross-view representation by utilizing the predicted correspondence, and the other is to perform data alignment in the latent space learned by the neural network.

## 3.2 Cross-view Representation Learning Module

To tackle the partially view-aligned problem in MVC, we propose using the aligned data $\{\mathbf{A}^{(v)}\}_{v=1}^{m}$ to train a neural network which is with the following objective function:

$$\mathcal{L} = \mathcal{L}_1 + \lambda \sum_{v \neq u} \mathcal{L}_2^{(uv)}, \tag{2}$$

where $\mathcal{L}_1$ is the loss for learning common representation across views, $\mathcal{L}_2$ is the loss between the model prediction and ground-truth correspondence. To be exact,

$$\mathcal{L}_1 = \sum_{v=1}^{m} \underbrace{\|\mathbf{A}^{(v)} - g^{(v)}(f^{(v)}(\mathbf{A}^{(v)}))\|_2^2}_{\text{within-view reconstruction}} + \sum_{v \neq u} \underbrace{\|f^{(v)}(\mathbf{A}^{(v)}) - \mathbf{P}_{uv}f^{(u)}(\hat{\mathbf{A}}^{(u)})\|_2^2}_{\text{cross-view consistency}}, \tag{3}$$

---

**Algorithm 1** Optimization of $\mathbf{P}$

---

   **Input:** $\mathbf{D}, \mathbf{P_0}, \sigma, \{\tau_i\}_{i=1}^2$
   **Initialize:** $\mathbf{P} = \mathbf{P_0}$
   **for** $i < \tau_1$ **do**
      $\mathbf{P} = \mathbf{P} - \sigma\mathbf{D}, \mathbf{T}_0 = \mathbf{P}, d_1 = d_2 = d_3 = 0$
      **for** $j < \tau_2$ **do**
         $\mathbf{T}_1 = \Psi_1(\mathbf{T}_0 + d_1), d_1 = \mathbf{T}_0 + d_1 - \mathbf{T}_1$
         $\mathbf{T}_2 = \Psi_2(\mathbf{T}_1 + d_2), d_2 = \mathbf{T}_1 + d_2 - \mathbf{T}_2$
         $\mathbf{T}_3 = \Psi_3(\mathbf{T}_2 + d_3), d_3 = \mathbf{T}_2 + d_3 - \mathbf{T}_3$
      **end for**
      $\mathbf{P} = \mathbf{T}_3$
   **end for**
   **Output:** $\mathbf{P}$

---

where the $v$-th autoencoder $\{f^{(v)}, g^{(v)}\}$ aims to learn a view-specific latent space for the $v$-th view by minimizing the reconstruction error. $\hat{\mathbf{A}}^{(u)}$ denotes the shuffled aligned data from $\mathbf{A}^{(u)}$ which is used to simulate the unaligned data. In brief, $\mathcal{L}_1$ consists of within-view reconstruction and cross-view consistency term which aims to learn the common representation by preserving the view-specific structure and cross-view consistency.

To exploit the alignment information in $\mathbf{A}^{(v)}$ to facilitate MVC, $\mathcal{L}_2^{(uv)}$ is defined to minimize the loss between the predicted and ground-truth alignment on the simulated unaligned data $\hat{\mathbf{A}}^{(u)}$ via

$$\mathcal{L}_2^{(uv)} = \|\mathbf{P}^{(uv)} - \mathbf{P}_{gt}^{(uv)}\|_2^2, \tag{4}$$

where $\mathbf{P}_{gt}^{(uv)}$ is the permutation ground truth from $\mathbf{A}^{(u)}$ w.r.t. $\mathbf{A}^{(v)}$. $\mathbf{P}^{(uv)}$ is the learned permutation matrix which establishes the correspondence between $\mathbf{A}^{(v)}$ and $\hat{\mathbf{A}}^{(u)}$. We experimentally adopt the $\ell_2$-norm loss term here, but other metrics such as Hamming distance could also be used. For ease of presentation, we take the binary-view data as a showcase by letting $m = 2$ without loss of generality. Note that, our model could easily extend to multiple views by selecting one view as the anchor, and align the other views to establish the correspondence with the corresponding permutation matrix.

It should be pointed out that directly learning $\mathbf{P}$ through minimizing $\mathcal{L}$ will face two problems. On one hand, the properties (*e.g.*, binary value) of $\mathbf{P}$ are hard to guarantee. On the other hand, our main goal is not to obtain $\mathbf{P}$ on the aligned data $\mathbf{A}$. Instead, it is highly expected to have a parametric model to handle the unaligned data $\mathbf{U}$. In the next part, we will elaborate on the optimization of $\mathbf{P}$ in the neural network.

### 3.3 Differentiable Alignment Module

To achieve data alignment, the optimization of $\mathbf{P}$ could be defined as an integer linear programming (ILP) problem which aims at achieving the best matching of bi-graph. Formally,

$$\begin{aligned} \underset{\mathbf{P}}{\arg\min} \quad & Tr(\mathbf{DP}^\top) \\ s.t. \quad & P_{ij} \in \{0, 1\}, \forall(i, j) \\ & \mathbf{P1} = \mathbf{1} \\ & \mathbf{P}^\top\mathbf{1} = \mathbf{1}, \end{aligned} \tag{5}$$

where $Tr()$ denotes the matrix trace and $\mathbf{D} \in \mathbb{R}^{n \times n}$ is the distance matrix in which $D_{ij}$ denotes the distance of assigning $i$ to $j$. In the paper, we define $\mathbf{D}$ as the pairwise distance between $\mathbf{A}_i^{(1)}$ and $\hat{\mathbf{A}}_j^{(2)}$ in the latent space, *i.e.*,

$$D_{ij} = \|f^{(1)}(\mathbf{A}_i^{(1)}) - f^{(2)}(\hat{\mathbf{A}}_j^{(2)})\|_2^2. \tag{6}$$

As Eq.(5) is NP-complete [12] and non-differentiable, it is impossible to plug it into a neural network as expected. Therefore, we relax the constraint of the binary matrix into real-valued permutation

matrix and Eq.(5) could be rewritten as:

$$\begin{aligned} \arg\min \quad & Tr(\mathbf{DP}^\top) \\ s.t. \quad & P_{ij} \geq 0, \forall(i,j) \\ & \mathbf{P1} = \mathbf{1} \\ & \mathbf{P}^\top \mathbf{1} = \mathbf{1}, \end{aligned} \quad (7)$$

where the relaxation is also consistent with the goal of MVC, *i.e.*, the exact point level alignment in multi-view data is unnecessary to the clustering task. Instead, the set level alignment is more desirable.

As the above loss could be solved by a differentiable gradient descent algorithm through updating the permutation matrix $\mathbf{P}$ with the negative gradients. The remained challenge is how to keep the above three constraints when updating the permutation matrix. One could observe that the optimized set is the intersection of the three closed convex sets from the constraints above.

To obtain the optimized permutation matrix, we adopt the Dykstra's projection algorithm [3] that computes the intersection of convex sets by iteratively projecting the updated permutation matrix $\mathbf{P}$ onto each of the convex sets, which has been proved effective in many applications [24, 20, 35]. Similar to [35] which adopts Dykstra's projection algorithm to solve the ILP problem, we project the permutation matrix into three constraint sets individually as follows:

$$\Psi_1(\mathbf{P}) = \text{ReLU}(\mathbf{P}). \quad (8)$$

$$\Psi_2(\mathbf{P}) = \mathbf{P} - \frac{1}{n}(\mathbf{P1} - \mathbf{1})\mathbf{1}^\top. \quad (9)$$

$$\Psi_3(\mathbf{P}) = \mathbf{P} - \frac{1}{n}\mathbf{1}(\mathbf{1}^\top\mathbf{P} - \mathbf{1}^\top). \quad (10)$$

Obviously, $\Psi_1$, $\Psi_2$, and $\Psi_3$ project the permutation matrix $\mathbf{P}$ into the three constraint sets, respectively. It should be pointed out that the Dykstra's projection algorithm is not the unique solution. Other algorithms such as Sinkhorn normalization [26] could be alternative as long as $\mathbf{P}$ satisfies the aforementioned constraints. The optimization process is summarized in Algorithm 1.

It is attractive that the alignment operation is fully differentiable and only involves matrix multiplication and addition/division. Therefore, we further recast Algorithm 1 as a differentiable alignment module which is pluggable to the neural network as shown in Fig. 2. Note that, the module does not involve the parameter optimization, and the computation of $\mathbf{P}$ will be very fast. It aligns the data with the computation complexity $\mathcal{O}(\tau_1\tau_2 n^2)$ ($n$ is the batch size for training), and allows the network to utilize the available correspondence information from partially aligned data in an end-to-end manner, as shown in Eq.4. In general, the alignment module is pluggable for any neural network such as DCCA/DCCAE by computing the pairwise distance on the learned representations and achieving the alignment by the differentiable alignment module. In addition, the structure and output of the module enjoy interpretability derived from the ILP problem and the Dykstra's optimization.

### 3.4 Implementation Details

As shown in Fig. 2, the proposed PVC method consists of two modules, *i.e.*, representation learning module and alignment module. For the representation learning, we adopt two standard autoencoders for two views and have presented the details of the network structure and implementation in the supplementary materials.

We perform the representation learning and aligning in a cyclic order as below:

- Step 1 (Representation learning): Pass "unaligned" data through the network, *i.e.*, $\{f^{(1)}, f^{(2)}, g^{(1)}, g^{(2)}\}$, yielding the hidden representations $\mathbf{Z}^{(1)} = f^{(1)}(\mathbf{A}^{(1)})$ and $\mathbf{Z}^{(2)} = f^{(2)}(\hat{\mathbf{A}}^{(2)})$. Then we calculate the distance matrix $\mathbf{D}$ in the latent space via Eq. 6.

- Step 2 (View Aligning): Perform alignment by feeding $\mathbf{D}$ to obtain the permutation matrix $\mathbf{P}$. Moreover, we experimentally set $\tau_1 = 30$ and $\tau_2 = 10$ to speed up the computation.

- Step 3: Compute the loss as defined in Eq. 2 and then update the network parameters and weights via back-propagation.

- Step 4: Repeat Steps 1–3 until convergence.

Once the network converged, we feed the whole dataset into the network which will perform alignment and infer the corresponding latent representations with the aligned data. After that, the view-specific representations are simply concatenated as the common representation which is further used for clustering by k-means like the traditional fashion [30, 25, 37].

## 4 Experiments

In this section, we evaluate the proposed PVC method on four widely-used multi-view datasets with the comparisons of nine multi-view clustering approaches. We implement PVC in PyTorch and carry all evaluations on a standard Ubuntu-18.04 OS with an NVIDIA 2080Ti GPU. Due to the space limitation, some analysis experiments and the technical details about our method and experiments are presented in the supplementary materials.

### 4.1 Experiment Setting

We carry our experiments on four popular multi-view datasets including: **Caltech101-20** [15, 25] which consists of 2,386 images of 20 subjects with two handcrafted features as two views. **Reuters** [8] which is a subset of the Reuters database. It consists of 3,000 samples from 6 classes, using German and Spanish as two views. **Scene-15**[5] which consists of 4,485 images distributed over 15 scene categories with two views. **Pascal Sentences** [7] which is selected from 2008 PASCAL development kit. It contains 1,000 images of 20 classes with corresponding text descriptions. More details could refer to the supplementary material.

Table 1: Clustering performance comparison on four challenging datasets.

| Aligned | Methods | Reuters | | | Caltech101-20 | | | Scene-15 | | | Pascal | | |
|---|---|---|---|---|---|---|---|---|---|---|---|---|---|
| | | ACC | F-mea | NMI | ACC | F-mea | NMI | ACC | F-mea | NMI | ACC | F-mea | NMI |
| Partially | CCA | 39.70 | 30.90 | 14.25 | **42.20** | **36.15** | **60.84** | 32.46 | 26.00 | 35.81 | 62.56 | 60.24 | **65.19** |
| | KCCA | **40.67** | 33.18 | **16.79** | 26.61 | 23.82 | 31.56 | 30.86 | 26.72 | 31.12 | 38.78 | 36.57 | 41.79 |
| | DCCA | 40.40 | **33.72** | 15.47 | 23.60 | 21.44 | 30.64 | 33.24 | 28.94 | 34.27 | 42.33 | 39.88 | 47.67 |
| | DCCAE | 36.77 | 29.65 | 14.07 | 31.68 | 26.22 | 38.61 | 30.28 | 27.11 | 33.31 | 43.89 | 41.87 | 50.76 |
| | LMSC | 37.43 | 32.72 | 12.31 | 26.03 | 20.27 | 35.65 | 22.16 | 20.00 | 16.46 | 50.89 | 48.78 | 53.70 |
| | MvC-DMF | 34.63 | 23.56 | 11.21 | 19.74 | 17.43 | 21.58 | 21.54 | 19.78 | 19.31 | 30.00 | 28.81 | 32.33 |
| | SwMC | 32.93 | 19.15 | 15.30 | 38.94 | 21.09 | 30.14 | 18.73 | 15.85 | 21.30 | 49.00 | 46.38 | 54.65 |
| | BMVC | 34.90 | 24.01 | 9.24 | 31.73 | 23.74 | 50.78 | **36.74** | **32.91** | **37.47** | 50.89 | 49.21 | 52.71 |
| | AE$^2$-Nets | 37.67 | 30.73 | 13.03 | 29.97 | 26.16 | 47.64 | 29.52 | 26.45 | 28.43 | **66.56** | **66.02** | 64.89 |
| Fully | CCA | 39.37 | 29.84 | 13.98 | 39.52 | 33.90 | 57.53 | 34.38 | 29.07 | 37.37 | 38.89 | 36.32 | 44.85 |
| | KCCA | 47.07 | 42.15 | 23.32 | 42.20 | 37.95 | 56.4 | 37.24 | 30.86 | 36.85 | 44.33 | 41.45 | 49.66 |
| | DCCA | 47.10 | 41.93 | 23.53 | 41.95 | 35.89 | 60.72 | 35.88 | 30.08 | 39.90 | 63.44 | 59.94 | 68.08 |
| | DCCAE | 48.10 | 42.33 | 24.57 | 44.17 | 42.11 | 60.83 | 36.68 | 30.11 | **40.56** | 64.89 | 62.27 | 67.89 |
| | LMSC | **48.40** | **45.03** | **27.47** | 31.56 | 22.18 | 32.17 | 33.60 | 31.09 | 32.98 | 60.44 | 56.99 | 64.50 |
| | MvC-DMF | 38.80 | 28.55 | 17.81 | **59.72** | **38.16** | 62.76 | 29.70 | 26.45 | 29.72 | 56.33 | 53.17 | 61.18 |
| | SwMC | 33.33 | 22.21 | 24.01 | 52.68 | 34.96 | 56.87 | 27.47 | 24.55 | 35.71 | 65.44 | 62.28 | **72.48** |
| | BMVC | 40.87 | 37.31 | 17.28 | 39.65 | 30.79 | **63.24** | **40.16** | **35.69** | 40.30 | 64.89 | 60.01 | 71.54 |
| | AE$^2$-Nets | 40.67 | 33.06 | 15.50 | 30.09 | 19.11 | 32.31 | 36.16 | 34.14 | 39.98 | **69.67** | **69.53** | 70.08 |
| Partially | PVC | **50.67** | **48.44** | **27.99** | 48.07 | 55.03 | 65.39 | 37.32 | 33.05 | 39.33 | 62.67 | 62.84 | 72.33 |

We compare our PVC with nine multi-view clustering approaches including: canonically correlated analysis (CCA)[29], kernel canonically correlated analysis (KCCA)[2], deep canonically correlated analysis (DCCA) [1], deep canonically correlated autoencoders (DCCAE) [30], Multi-View Clustering via Deep Matrix Factorization(MvC-DMF) [40], latent multi-view subspace clustering (LMSC) [36], self-weighted multi-view clustering (SwMC) [23], binary multi-view clustering (BMVC) [39], and Autoencoder in Autoencoder Networks (AE$^2$-Nets) [37]. For all methods, we adopt the recommended network structure and parameters. In brief, for the CCA-based methods, we fix the hidden representation dimension to 10. For BMVC, we fix the length of binary code to 128. For LMSC, we fix the latent representation dimension to 100 and seek the optimal $\lambda$ from (0.01, 0.1, 1, 10). For MvC-DMF, we seek the optimal $\beta$ and $\gamma$ from (0.1, 1, 10, 100) as suggested.

For a comprehensive analysis, we adopt accuracy (ACC), normalized mutual information (NMI), and F-measure (F-mea) to evaluate all the tested methods. A higher value of these metrics indicates a better performance.

## 4.2 Comparison with State of The Arts

To evaluate the effectiveness of PVC on the partially view-aligned data, we first construct the partially view-aligned data from the above four datasets. For Caltech101-20, Reuters and Scene-15, we randomly split them into two partitions ($\{\mathbf{A}^{(v)}, \mathbf{U}^{(v)}\}_{v=1}^{m}$) with the equal size. The partition $\{\mathbf{A}^{(v)}\}_v$ remains the known correspondence and the partition $\{\mathbf{U}^{(v)}\}_v$ are randomly permuted. For the Pascal, we directly use the training set as $\{\mathbf{A}^{(v)}\}_v$ and shuffle the testing set as $\{\mathbf{U}^{(v)}\}_v$.

As the almost all existing MVC methods including the above baselines cannot handle with the partially view-aligned data, we adopt two alternative solutions for comparisons: 1) We adopt PCA to project the raw data into a latent space so that the Hungarian algorithm could be applied to establish the correspondence of $\{\mathbf{U}^{(v)}\}_v$. After that, we carry out these baselines on the aligned data to achieve clustering. This evaluation could demonstrate the effectiveness of our whole model. 2) We run these baselines on the original data which are with the ground-truth correspondence. In other words, these baselines directly handle the fully-aligned data without any preprocessing. This evaluation could verify our claim that set-level alignment could favor clustering performance.

Table 1 shows the quantitative comparison results, from which one could observe that: 1) In the first evaluation, our PVC remarkably outperforms the other methods by a considerable margin. Specifically, in terms of NMI, PVC achieves about $11.2\%$ (Reuters), $4.55\%$ (Caltech101-20), $1.86\%$ (Scene-15) and $7.14\%$ (Pascal) progress compared to the best baseline. 2) In the second evaluation, our method PVC still achieves competitive results even though the baselines are with ground-truth alignment whereas our method does not. It is surprising that PVC could be better than all baselines in this test on Reuters. The possible reason is two-fold. First, Reuters is a document database that consists of English documents and its machine-translated versions. Thus there may be some noise/incorrect pairs in the fully-aligned dataset which could be addressed by our alignment module. Second, the re-aligned multi-view data may improve data consistency and meet the intrinsic data distribution, thus boosting the clustering performance.

## 4.3 Ablation Studies and Parameter Analysis

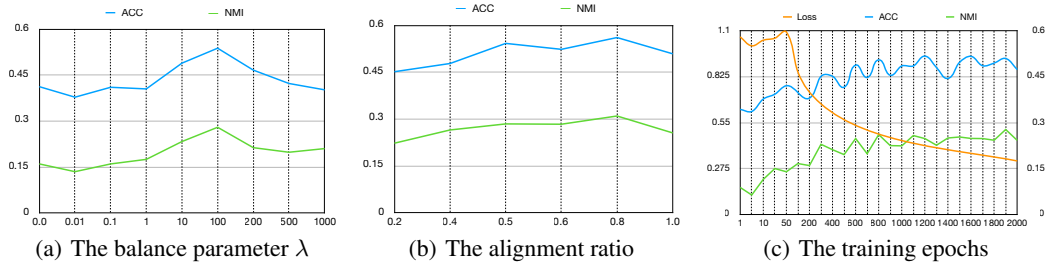

(a) The balance parameter $\lambda$     (b) The alignment ratio     (c) The training epochs

Figure 3: (a) Clustering performance of PVC with varying $\lambda$ on Reuters. (b) Clustering performance of PVC on the Reuters dataset with varying alignment ratio. (c) Clustering Performance of PVC with increasing epoch on the Reuters. The $x$-axis denotes the training epoch, the left and right $y$-axis denote the loss value and the corresponding clustering performance respectively.

To further investigate the influence of the parameter $\lambda$ of our method, we conduct the experiment on the Reuters dataset by reporting the ACC and NMI score with varying lambda values. Note that, the alignment module is ablated from our method when $\lambda = 0$. Fig. 3(a) demonstrates that ACC and NMI keep increasing until $\lambda = 100$ and then decline with increasing value.

Moreover, to investigate the performance of the proposed model on the partially aligned data with different unaligned proportions, we conduct experiments on the Reuters dataset. From Fig. 3(b), one could observe that the clustering performance generally increases with the alignment ratio. Note that, PVC achieves the best result when $80\%$ rather than $100\%$ of data are with ground-truth correspondence. The reason may come from our alignment module. Specifically, in the testing stage, all data will be realigned and thus lead to an inferior result.

### 4.4 Convergence Analysis

In this section, we investigate the convergence of our PVC by reporting the loss value and the corresponding clustering performance with increasing epoch. As shown in Fig. 3(c), one could observe that the loss decreases a lot in the first 600 epoch, then continuously decrease until convergence. As for the clustering performance, both the ACC and NMI continuously increases in the first 600 epoch and then only keep fluctuation in a narrow range.

## 5 Conclusion

In this paper, a challenging problem in multi-view clustering, namely the partially view-aligned problem, is studied for the first time. The solution to this problem could alleviate intensive labor for fully-aligned data collection. To solve this challenging problem, we propose a novel multi-view clustering method termed as partially view-aligned clustering consists of a differentiable alignment module and a representation learning module. The alignment module is a differentiable surrogate of the non-differentiable Hungarian algorithm, which could establish the correspondence of two views. Besides, the module could enjoy the high interpretability in neural structure and outputs as it is derived from the Dysktra's projection algorithm. Extensive experiments verify the effectiveness of our learning paradigm. In the future, we plan to explore the potential of our method to handle other multi-view analysis tasks. Moreover, it is still unknown how to handle fully unaligned data and the data which simultaneously encounters the missing views and unaligned data problems.

## Broader Impact Statement

Multi-view clustering is a common topic in multi-view learning which could be applied to a wide range of applications including computer vision, recommender systems, data retrieval, natural language processing. Our work could address the partially view-aligned problem faced by many real-world applications and perform multi-view clustering.

While there will be important impacts resulting from the use of PVC in general (depend on various multi-view applications), here we focus on the impact of using our method to address the partially view-aligned problem which is widely faced by the real-world applications. There are many benefits to solving this problem, such as reducing the costs of manually aligning multi-view data, increasing the robustness for downstream tasks by handling PVP. Besides the benefits we should also care about the potential negative impacts including 1) The risk of automation bias [21] for decision making, especially in aviation, health care, and autonomous vehicles. 2) The job loss caused by the PVC since it can automatically establish the correspondence on the unaligned data. Usually, this requires domain experts to manually align them.

We would encourage further work to understand and mitigate the above biases and risks. Concerning the risk of automation bias, we encourage research to understand the final decision with domain expertise.

## Acknowledgements

This work was supported in part by the National Key R&D Program of China under Grant 2020YFB1406702; in part by NFSC under Grant U19A2081, 61625204, and 61836006; in part by the Fundamental Research Funds for the Central Universities under Grant YJ201949; and in part by A*STAR AME Programmatic under Grant A18A1b0045.

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
