[Supplementary Material]

# Supplementary Material for
# Partially View-aligned Clustering

**Zhenyu Huang**
College of Computer Science
Sichuan University, China
zyhuang.gm@gmail.com

**Peng Hu**
I2R
A*STAR, Singapore
penghu.ml@gmail.com

**Joey Tianyi Zhou**
iHPC
A*STAR, Singapore
joey.tianyi.zhou@gmail.com

**Jiancheng Lv**
College of Computer Science
Sichuan University, China
lvjiancheng@scu.edu.cn

**Xi Peng**$^*$
College of Computer Science
Sichuan University, China
pengx.gm@gmail.com

## 1 Introduction

In this supplementary material, we provide additional information including the network architecture, the training configuration, and implementation details. To investigate the effectiveness of the proposed differentiable aligning module, we conduct the experiments on the Pascal dataset to obtain the aligning results by Hungarian and our method. Furthermore, in the main paper, we report the performance comparison on the partially aligned data with the Hungarian algorithm performed on the PCA projected data. For a more complete comparison, in this material, we replace the PCA technology with auto-encoders (AEs) and report the results of the multi-view clustering baselines on the partially aligned data.

## 2 Experiment Details

In this section, we elaborate on the dataset description, the implementation details of our method, and the experimental settings.

### 2.1 Dataset

We introduce the four used popular multi-view datasets as follows:

- **Caltech101-20** [8, 12]: The dataset consists of 2,386 images of 20 subjects with six hand-crafted features as six views, including Gabor feature, Wavelet Moments, CENTRIST feature, HOG feature, GIST feature and LBP feature. We use the HOG and GIST features in the following experiments.

- **Reuters** [6]: We use a subset of the Reuters database which consists of 18,758 samples from 6 classes. The dataset is constructed with five languages as different views including the English version and the translations in four different languages, i.e., French, German, Spanish and Italian. We adopt the standard autoencoder to project the raw data into 10-dim representations similar to [6] and use only 3,000 samples in German and Spanish.

- **Scene-15**: The dataset [4] consists of 4,485 images distributed over 15 indoor and outdoor scene categories. Three image features are used as views similar to [3], i.e., GIST, PHOG, and LBP. We only use the PHOG and GIST features in the following experiments.

---

$^*$Corresponding author.

- **Pascal Sentences** [5]: This dataset is selected from 2008 PASCAL development kit, which contains 1,000 images of 20 classes with corresponding text descriptions. We follow the same setting in [5]. The Pascal Sentences dataset is split into three parts including the training set with 800 pairs, the testing set with 100 pairs, and the validation set with 100 pairs. Note that here we only use the training set and testing set with totally 900 samples.

## 2.2 Implementation Details of PVC

**Network Design:** The proposed method contains two modules including the representation learning and alignment module. We have shown the technical detail about the alignment module (see Fig. 2 and Algorithm 1 in the main paper). Here we give the detail information of the network architectures in the representation learning module which has two auto-encoders for two views. Table 1 gives the detail architecture of the AEs which only use the dense layer (i.e. full-connected) with the ReLU activation function.

**Training Details:** We implement the proposed model in PyTorch [10]. For all experiments, we use the Adam [7] optimizer with the default learning rate (LR) of $10^{-3}$ with weight decay $10^{-5}$. Before the training, we first pretrain the network for fast convergence for a specified number of epochs (denoted as pretrain epoch) with loss $\mathcal{L}_1$, i.e. without the alignment module. Then we train the network with loss defined in the main paper (see Eq. 2) with a balance factor $\lambda$ for a specified number of epochs (denoted as training epoch). More training details about PVC are presented in Table 2. Note that we experimentally select the $\lambda$ in range $\{0.001, 0.01, 0.1, 1, 10, 100, 1000\}$ and report the best. The batch size is selected experimentally according to the dataset size. And we experimentally set the pretrain epoch and training epoch according to the convergence.

Table 1: The architecture of the used auto-encoders in PVC. Here we only give the architectures of the encoders, the decoders consist of same layers with reverse order.

| Dataset | Encoder |
|---|---|
| Reuters | Dense (ReLU, size = 1024) |
| | Dense (ReLU, size = 1024) |
| | Dense (ReLU, size = 1024) |
| | Dense (ReLU, size = 10) |
| Caltech101-20 | Dense (ReLU, size = 1024) |
| | Dense (ReLU, size = 1024) |
| | Dense (ReLU, size = 20) |
| Scene-15 | Dense (ReLU, size = 2048) |
| | Dense (ReLU, size = 1024) |
| | Dense (ReLU, size = 512) |
| | Dense (ReLU, size = 128) |
| Pascal | Dense (ReLU, size = 1024) |
| | Dense (ReLU, size = 512) |
| | Dense (ReLU, size = 10) |

Table 2: The parameter settings of training on four datasets.

| Dataset | $\lambda$ | Batch | LR | Weight decay | Optimizer | Pretrain Epochs | Training Epochs |
|---|---|---|---|---|---|---|---|
| Reuters | 100 | 128 | $10^{-3}$ | $10^{-5}$ | Adam | 1000 | 2000 |
| Caltech101-20 | 10 | 128 | $10^{-3}$ | $10^{-5}$ | Adam | 2000 | 2000 |
| Scene-15 | 100 | 64 | $10^{-3}$ | $10^{-5}$ | Adam | 500 | 2000 |
| Pascal | 3 | 32 | $10^{-3}$ | $10^{-5}$ | Adam | 500 | 2000 |

## 2.3 Clustering Implementation Details

To perform the clustering on the obtained representation, we adopt the $k$-means to compute the cluster assignments on the final representation which is the concatenation of the hidden representations from different views. Specifically, we use the $k$-means contained into the Scikit-Learn package [11] with the default configuration. For a fair comparison, we repeat $k$-means 10 times with different

Table 3: The architectures the auto-encoder.

| Encoder | Decoder |
|---|---|
| Dense (ReLU, size = 500) | Dense (ReLU, size = 10) |
| Dense (ReLU, size = 500) | Dense (ReLU, size = 2000) |
| Dense (ReLU, size = 2000) | Dense (ReLU, size = 500) |
| Dense (tanh, size = 10) | Dense (ReLU, size = 500) |

Table 4: Clustering performance comparison on four challenging datasets.

| Aligned | Methods | Reuters | | | Caltech101-20 | | | Scene-15 | | | Pascal | | |
|---|---|---|---|---|---|---|---|---|---|---|---|---|---|
| | | ACC | F-mea | NMI | ACC | F-mea | NMI | ACC | F-mea | NMI | ACC | F-mea | NMI |
| Partially | CCA | 39.70 | 30.90 | 14.25 | **39.44** | **35.68** | 56.54 | 31.86 | 28.19 | **35.49** | 51.89 | 49.06 | 54.75 |
| | KCCA | **40.67** | 33.18 | **16.79** | 26.49 | 21.77 | 30.67 | 25.98 | 23.27 | 28.45 | 50.33 | 46.43 | 54.66 |
| | DCCA | 40.40 | **33.72** | 15.47 | 25.44 | 19.21 | 30.42 | 25.64 | 20.25 | 23.47 | 49.56 | 45.95 | 54.70 |
| | DCCAE | 36.77 | 29.65 | 14.07 | 30.22 | 23.66 | 36.47 | 24.62 | 19.78 | 22.54 | 53.00 | 50.21 | 58.86 |
| | LMSC | 37.43 | 32.72 | 12.31 | 27.33 | 21.72 | 44.41 | 23.43 | 21.65 | 24.22 | 60.56 | 57.47 | 59.46 |
| | MvC-DMF | 34.63 | 23.56 | 11.21 | 26.78 | 18.01 | 23.55 | 16.97 | 16.01 | 11.60 | 37.56 | 35.59 | 41.63 |
| | SwMC | 32.93 | 19.15 | 15.30 | 32.86 | 22.09 | 27.92 | 12.98 | 7.91 | 12.80 | 49.67 | 47.00 | 55.34 |
| | BMVC | 34.90 | 24.01 | 9.24 | 31.48 | 23.59 | **56.65** | **33.11** | **30.75** | 31.39 | 55.00 | 52.17 | 61.17 |
| | AE$^2$-Nets | 37.67 | 30.73 | 13.03 | 30.60 | 22.58 | 44.87 | 25.31 | 24.44 | 29.28 | **60.89** | **60.41** | **61.27** |
| Partially | PVC | **50.67** | **48.44** | **27.99** | **48.07** | **55.03** | **65.39** | **37.32** | **33.05** | **39.33** | **62.67** | **62.84** | **72.33** |

centroid seeds and report the best results. In all experiments, we adopt three metrics implemented by Scikit-Learn to evaluate the clustering performance, namely, ACC, F-measure, NMI.

## 2.4 Hungarian vs. Differentiable Alignment Module

(a)                                (b)                                (c)

Figure 1: The comparison of the Hungarian algorithm and our differentiable alignment module. The results are obtained on the Pascal dataset. (a) The permutation matrix obtained by the Hungarian algorithm on the raw data of which all views are with the same dimension via PCA; (b) The predicted real-value permutation matrix obtained by PVC; (c) The predicted binary permutation matrix obtained by PVC. We indicate the blocks with the green boxes and arrange the data cluster by cluster for better visualization and each block corresponds to a cluster.

In this section, we conduct analysis of our differentiable alignment module compared with the Hungarian algorithm from the qualitative and quantitative perspectives.

We firstly investigate the learned permutation matrix through visualization. To be specific, we conduct experiments on the Pascal dataset with the first 400 samples with 10 classes for visualization clarity. As shown in Fig. 1, three learned permutation matrixes are shown. Fig. 1(a) shows that the Hungarian algorithm obtains an undesirable permutation matrix which is in disarray. Fig. 1(b)–1(c) show the

block-diagonal structure which indicates that the set-level alignment information is well preserved by our method, where Fig. 1(c) is obtained by setting the maximum of each row of Fig. 1(b) to 1 and 0 elsewhere.

Table 5: Quantitative comparison between the Hungarian method and our differentiable alignment module.

| Method | Alignment Errors | Time |
|---|---|---|
| Hungarian | 366 | 17.78s |
| Ours (w.o. $\mathcal{L}_2$ ) | 316 | 0.09s |
| Ours (w. $\mathcal{L}_2$ ) | 132 | 0.08s |

Besides the above qualitative analysis, we also quantitatively compare our alignment method with the Hungarian algorithm in terms of the alignment errors and time cost. The alignment error is the number of between-cluster correspondence. As shown in Table 5, our method performs remarkably better than the Hungarian in terms of alignment accuracy and time cost. Note that, the time complexity of the fastest Hungarian algorithm is proportional to the cube of the data size (*i.e.*, $O(n^3)$), whereas our alignment module only involves matrix multiplication and addition/division in the operation of $O(n^2)$. Note that, the Hungarian algorithm runs on the CPU, whereas our method runs on the GPU. Here, the time cost of them is just for a direct comparison.

In Table 5, we also reported the alignment error of our method without the differentiable alignment module, denoted by "Ours (w.o. $\mathcal{L}_2$)", by letting $\lambda = 0$.

### 2.5   Experiment with AEs

In the main body of our submission (Section 4.2), we report the results on the partially aligned data compared with the multi-view clustering aprroaches including: CCA[13], KCCA[2], DCCA [1], DCCAE [14], MvC-DMF [18], LMSC [15], SwMC[9], BMVC [17], and AE$^2$-Nets [16]. Specifically, we adopt two alternative solutions for comparisons: 1) We adopt PCA to project the raw data into a latent space so that the Hungarian algorithm could be applied to establish the correspondence of $\{\mathbf{U}^{(v)}\}_v$. Then, we carry out these baselines on the aligned data. 2) We run these baselines on the original data which are with the ground-truth correspondence.

For a more complete comparison, here we adopt the AEs shown in Table 3 to project the raw data into a latent space and the Hungarian algorithm is applied to establish the correspondence of $\{\mathbf{U}^{(v)}\}_v$ which is similar to the first solution above. After that, we carry out the above baselines on the aligned data to achieve clustering with the aforementioned setting in the main paper. Note that Reuters (document dataset) has features with the same dimension across different views as we already perform AEs to obtain the features as mentioned in the dataset introduction (see Section 4.1 in the main paper). Thus the results on Reuters are the same as the results in the main paper (see Table. 1 in the main paper).

As shown in Table 4, PVC remarkably outperforms the other methods by a considerable margin. Specifically, in terms of NMI, PVC achieves 11.2% (Reuters), 8.74% (Caltech101-20), 3.84% (Scene-15) and 11.06% (Pascal) progress compared to the best baseline.