[Reviews · NeurIPS 2020]

Review 1

Summary and Contributions: (1) A partially view-aligned problem (PVP) is studied in this paper. This is an interesting and novel problem which has been ignored by existing multi-view learning methods. (2) A partially view-aligned clustering method is proposed to solve the PVP, which contains a differentiable data alignment neural network.

Strengths: The PVP is a new setting with broad application and it is not enough investigated in the community. In addition, I think that it is with some practical values. For example, when one accesses to the data from multiple cameras, different cameras would provide a partial view of the same place due to difference receptive field, and aligning them and reasoning about it seems difficult and interesting. The proposed representation learning module and the differentiable alignment model are reasonable to improve the feature representation capability and solve the problem of partially aligned views. Experimental results also demonstrate the effectiveness of the model.

Weaknesses: The discussion on the partially view-aligned problem is not comprehensive enough to me. Besides the multi-view community, do the other communities such multi-modality will face this issue? Has it studied in these communities? If did, what the difference with them? This problem seems quite important and it is unclear why previous works ignored it. More literature review is needed. Some important details of the network setting is missing. What is the memory cost and parameter complexity of the proposed model? How about the compared AE2-Nets?

Correctness: Yes

Clarity: It is easy to read, but further revision is needed.

Relation to Prior Work: More discussion is needed.

Reproducibility: Yes

Additional Feedback: 1. Besides the multi-view community, do the other communities such multi-modality will face this issue? Has it studied in these communities? If did, what the difference with them? 2. What specific characteristics of clustering are utilized in the proposed PVC method? 3. Is it ok to plug the proposed differentiable aligning module into other neural network? How about the convergence of the module?


Review 2

Summary and Contributions: The paper studies a special Multi-view clustering problem in which the data in different views might be partially aligned, i.e., the columns across different views might come from different categories. In the proposed approach, the neural network learns the permutation matrix in an end-to-end manner. 

Strengths: 1. This paper investigates the special Muti-view clustering problem in which the data in different views might be partially aligned, i.e., the columns across different views might come from different categories. Such a so-called PVP is novel to the community. 2. To solve the view alignment, PVP is reformulated as an integer linear programming problem which is further solved in a differentiable way so that the learning of the permutation matrix could jointly work with representation learning of deep neural networks.

Weaknesses: 1. Why use ReLu in eq.8; 2. Is the differentiable module (eq8-10, Fig2.b) work in a recurrent manner? If does, how many layers (iteration) in the module. How about the convergence of this module, as well as its influence on the entire network. 3. Why in some tests, your method (with partially aligned data) could be better than the methods with fully aligned data?

Correctness: Yes

Clarity: Yes

Relation to Prior Work: Yes

Reproducibility: Yes

Additional Feedback: 1.Why use ReLu in eq.8; 2.Is the differentiable module (eq8-10, Fig2.b) work in a recurrent manner? If does, how many layers (iteration) in the module. How about the convergence of this module, as well as its influence on the entire network. 3.Why in some tests, your method (with partially aligned data) could be better than the methods with fully aligned data?


Review 3

Summary and Contributions: The work studies partially view-aligned problem, i.e., for two data matrices X1 and X2 corresponding to two views, they do not assume that X1 and X2 are fully aligned in row-wise. Instead, only a small portion of the matrices has established the correspondence in advance. To solve this problem, a novel multi-view clustering method termed as partially view-aligned clustering is proposed, which consists of a differentiable alignment module and a representation learning module. Extensive experiments verify the effectiveness of our learning paradigm.

Strengths: 1. The partially view-aligned problem is novel as far as I known; 2. The proposed solution achieves promising result, and it seems also novel and effective to learn the data permutation matrix in a differentiable neural module. 3. The paper is well-written and easily followed, and extensive experiments are conducted.

Weaknesses: 1. I like the solution of learning the permutation matrix in a neural network, however, some details are expected to this module, esp, could the module be trained in an end-to-end manner? If does, as this module seems like a recurrent network, which one optimizer is used to train your whole network? The convergence analysis should also be conducted in an ablation studies. 2. Why adopt the relu in your module? I do not see the detailed derivation, any specific reason? Are there other choices?

Correctness: Yes

Clarity: Yes

Relation to Prior Work: Yes

Reproducibility: Yes

Additional Feedback: 1. I like the solution of learning the permutation matrix in a neural network, however, some details are expected to this module, esp, could the module be trained in an end-to-end manner? If does, as this module seems like a recurrent network, which one optimizer is used to train your whole network? The convergence analysis should also be conducted in an ablation studies. 2. Why adopt the relu in your module? I do not see the detailed derivation, any specific reason? Are there other choices?


Review 4

Summary and Contributions: This paper studies partially view-aligned clustering, which is a new problem and has not been explored before. Giving two datasets from two modalities that are not fully aligned, the proposed method reconstructs the data in a latent space and perform an alternate of Hungarian algorithm to generate alignment. Comprehensive evaluations are conducted on multiple benchmark datasets.

Strengths: 1. This paper presents a new paradigm of multi-view learning, i.e., partially view-aligned clustering. Considering the difficulty of full data alignment in practice, this paper would inspire many other research topics in machine learning. 2. The proposed model could effectively utilize the partially-aligned data for model training. 3. The proposed differentiable alignment module is a surrogate of the non-differentiable Hungarian algorithm. This new module could be incorporated into many other machine learning models.

Weaknesses: 1. In experiments, Table 1 shows that partially aligned PVC obtains even better than baselines with full alignment. It seems that, in the partially aligned setting, A and U have the same size. The authors may add discussions on the performance of PVC when U is much larger than A. 2. I wonder if the model could handle multi-view data. As the proposed method matches two graphs corresponding to two views, it seems difficult to extend the method to multiple graphs. 3. In Fig.3, why the convergence curve is fluctuant with the training epoch? 4. Some typos in the current version should be addressed.

Correctness: The claims and proposed method in this paper are correct. The empirical methodology is also correct.

Clarity: The paper is very well written. Technical details are easy to follow.

Relation to Prior Work: This paper presents a new problem setting. The authors provide comprehensive comparisons between this paper and prior works.

Reproducibility: Yes

Additional Feedback: Update: the authors' response has addressed my previous concerns. Thus, I'd like to keep my original score.

[Author Response · NeurIPS 2020]

**R2 & R3 - Why use ReLU:** We use ReLU ($f(x) = \max(0, x)$) to enforce $\mathbf{P}$ positive. As the solution to integer linear programming (ILP) problem in Eq.5 is non-differentiable and NP-complete, we relax this constraint to Eq.6 through ReLU. After that, we adopt the Dykstra's projection algorithm to compute the intersection of convex sets by iteratively projecting $\mathbf{P}$ onto each of the convex sets, i.e., 1) $P_{ij} \geq 0$; 2) $\mathbf{P1} = \mathbf{1}$; 3) $\mathbf{P}^\top \mathbf{1} = \mathbf{1}$. In summary, we adopt ReLU to meet the first constraint, and the other functions will be alternative as long as it could make $\mathbf{P}$ positive.

**R1 & R2 & R3 - Alignment module:** The proposed module aligns the data with the computation complexity $\mathcal{O}(\tau_1 \tau_2 n^2)$ ($n$ is the batch size for training), and allows the network to utilize the available correspondence information from partially aligned data in an end-to-end manner, as shown in Eq.4. We use Adam optimizer for network training. The alignment module recurrently computes the permutation matrix, but it is not the RNN architecture. The proposed module is pluggable for any neural network such as DCCA/DCCAE by computing the pairwise distance on the learned representations and achieving the alignment by the proposed module.

**R1 & R2 & R3 & R4 - Convergence analysis:** The convergence of the whole model could refer to Fig3.c. One could observe that the loss decreases a lot in the first 600 epochs, then continuously and smoothly decreases until convergence. As for the alignment module, we experimentally find that it converges fast with ($\tau_1 = 30, \tau_2 = 10$). The influence of the alignment module to the whole network is predictable, since PVC jointly learns the common representation and aligns the data to address the challenging PVP, it inevitably converges slower than representation learning only.

**R2 & R4 - Results compared to fully aligned:** The possible reason is two-fold. First, Reuters is a document database that consists of English documents and its machine-translated versions. Thus there may be some noise/incorrect pairs in the fully-aligned dataset which could be addressed by our alignment module. Second, the re-aligned multi-view data may improve data consistency and meet the intrinsic data distribution, thus boosting the clustering performance.

**R1 - Literature review:** 1) Multi-modality also faces the partially view-aligned problem (PVP) as different modality may be collected in the wrong order due to temporal and spatial complexity. 2) In the paper, L85-95 have indicated that only a few works try to alleviate the effect caused by PVP. The major reasons for hindering studies on PVP have been stated in L107-117. In short, the traditional shallow methods usually pre-align the data in the preprocessing phrase and then perform clustering on the re-aligned data with a two-stage paradigm, which is to avoid directly solving PVP. In other words, PVP is ignored in the traditional shallow setting which does not benefit from end-to-end optimization. Moreover, the non-differentiable alignment algorithms adopted by these methods hinder them extend to deep models, while the proposed differentiable alignment module is pluggable to multi-view models to address PVP and embrace the attributes of deep models. 3) The difference between PVC and the existing works is two-fold. First, the methods are shallow models and there are no efforts devoted to developing effective deep solutions so far as we knew, while PVC proposes the differentiable alignment module to facilitate the deep approach. Second, these works establish the correspondence of views in a separate step, while PVC jointly learns the common representations and aligns the data.

**R1 - Network setting, memory cost, and parameter complexity:** We provide the configuration and implementation details of PVC in the supplementary material. For the memory cost and parameter complexity, we conduct experiments on Caltech101-20 compared to AE2-Nets. For training, PVC occupies 1126 MiB GPU memory and needs about 1.02 hours to convergence, while AE2-Nets occupies 362 MiB and needs 0.62 hours to convergence. The reason why PVC needs more memory and computation cost is the additional memory and computation cost caused by the alignment module, which is to address the challenging PVP. For testing, Table 5 (Supplementary Materials) gives a comparison of the Hungarian and the alignment module. It shows that the proposed alignment module (0.09s) is much faster than the Hungarian (17.78s), which means PVC is more capable of practical applications when the model is well-trained.

**R1 - Clustering characteristics:** As presented at L20-L22, most existing multi-view clustering approaches jointly learn a common representation to bridge the gap among different views and then achieve clustering on the common representation. In other words, learning the common representation is the key problem for multi-view clustering. Similarly, PVC jointly learns the common representation while enforcing the cross-view consistency with the help of the re-aligned data by the differentiable alignment module.

**R4 - When U larger than A:** Fig.5 may be helpful to address this concern. The figure shows that our method achieves a promising result (ACC: 0.4517, NMI: 0.2231) when U (=0.8) is remarkably larger than A (=0.2), showing the superiority of PVC even there are more unaligned data than aligned ones.

**R4 - Multiple views:** Our model could easily extend to multiple views by selecting one view as the anchor, and align the other views to establish the correspondence with the corresponding permutation matrix.

**R4 - Fluctuant curve:** From Fig.3.c, one could see that PVC loss decreases a lot in the first 600 epochs, then continuously and smoothly decreases until convergence. As for ACC and NMI, they both increase roughly as the epoch increase with the fluctuant curve. The possible reason for the fluctuant curve is that the re-aligned data may contain noise/incorrect correspondence, thus leading to unstable ACC and NMI.

[Meta-Review · NeurIPS 2020]

All reviewers fully support acceptance of this paper and I would also like to recommend acceptance. In particular all reviewers point that PVP is a novel problem not currently addressed by exiting multi-view learning methods, R1 and R3 comment that the results are promising and R1 and R4 point that the paper has broader applications that can inspire other work in the topic.